# Numerical Investigation of Triaxial Shear Behaviors of Cemented Sands with Different Sampling Conditions Using Discrete Element Method

**DOI:** 10.3390/ma15093337

**Published:** 2022-05-06

**Authors:** Xuqun Zhang, Zhaofeng Li, Pei Tai, Qing Zeng, Qishan Bai

**Affiliations:** 1Guangzhou Metro Design & Research Institute Co., Ltd., Guangzhou 510080, China; zhangxuqun@dtsjy.com; 2School of Civil and Environmental Engineering, Harbin Institute of Technology, Shenzhen 518055, China; taipei@hit.edu.cn (P.T.); zengqing@hit.edu.cn (Q.Z.); bqs666@163.com (Q.B.)

**Keywords:** cemented sand, discrete element method, triaxial shear test, bond breakage, shear band

## Abstract

In cemented sand, the influences of the sampling factors (i.e., the curing time, cement–sand ratio, and initial void ratio) on the triaxial shear behavior were investigated using discrete element method. Cemented sand samples with different initial conditions were prepared and subjected to the consolidated drained triaxial shearing test. In the simulations, the peak strength, residual strength, and pre-peak stiffness of cemented sand were enhanced by increasing the curing time and cement–sand ratio, and the enhancements could be explained by the increases in bond strength and bond number. Resulting from the increases of these two sampling factors, bond breakage emerged at a greater axial strain but lower intensity. However, some uncommon phenomena were generated; that is, the contractive but strain-softening response occurred in the sample with a curing time of 3 days, and the shear band and the strain-hardening behavior coexisted in the sample with a cement–sand ratio of 1%. The peak strength and pre-peak stiffness were also enhanced by decreasing the initial void ratio, more distinctly than by increasing the curing time and cement–sand ratio. However, the residual strength, bond breakage, and failure pattern with the persistence of shear band were insensitive to this change.

## 1. Introduction

In engineering practice, the strength of loose sand can be efficiently enhanced by adding a cementation agent, such as Portland cement, gypsum, and calcite precipitated microbially [1,2,3,4]. The loose sand is then turned into the cemented sand, with bonds usually formed at the contacts between sand particles. Therefore, to provide reliable instructions for sand stabilization by cementation, the behavior of cemented sand should be well understood.

Laboratory tests on cemented sand have shown that the behavior of cemented sand is sensitive to the sampling condition, thus presenting difficulties when developing the associated constitutive model [5,6,7] and evaluating the performance of cement-stabilized soil in ground improvement [8,9]. Consoli et al. [10] suggested the curing time as one of the influential factors for the strength of cemented sand. Li et al. [11] revealed the importance of the cement–sand ratio (i.e., the mass of cement divided by the mass of dry sand) or cement content to the stress–strain response of cemented sand, and, in their tests, the strain-softening behavior was produced when only 1% cement was used in sample preparation. The significance of the initial void ratio was also recognized, and a variable named the cement–void ratio (i.e., the porosity over the cement–sand ratio) was further proposed and found to be positively correlated with strength and stiffness [12,13]. According to these laboratory tests, the sampling factors of the curing time, cement–sand ratio, and initial void ratio appear to play a major role in the mechanical responses of cemented sand, while other factors, such as the curing stress and particle size, may be correlated with the three major ones and only play a minor role [14,15,16].

The above experimental studies have helped to identify the sampling factors which are most significant for cemented sand. However, these studies mainly focused only on the peak strength of cement sand. In fact, to establish the constitutive model, other properties such as the stiffness, residual strength, and volumetric strain are also needed [17,18]. Furthermore, merely by experimentation, these works are insufficient to unravel the regulation mechanisms of the sampling factors toward the soil responses, which are related to the bond breakage at the microscale and are also important for constitutive modeling [19,20,21,22]. Therefore, in parallel with laboratory tests, simulations using the discrete element method (DEM) have been frequently carried out in the study of cemented sand [23,24,25,26]. This is because DEM simulations, on one hand, can provide microscopic insight into the soil behavior [27,28,29] and, on the other hand, can control the sampling and shearing process in an accurate and idealized manner, eliminating the bias from manual operation [30,31,32,33,34]. Using this technique, Li et al. [26] preliminary discussed the influence of curing time (essentially the bond strength) on the stress–strain response of cemented sand; that is, the strain-softening response and the shear band emerged in the long-term cured sample, while the strain-hardening response and the bulging-type failure pattern emerged in the short-term cured sample. Furthermore, the evolution of bond breakage was captured to explain the influence of curing time, which could not be achieved in the laboratory test. However, for other important sampling factors, i.e., the cement–sand ratio and initial void ratio, such a numerical investigation has still not been carried out.

Therefore, in this study, the influences of the sampling factors (i.e., the curing time/bond strength, cement–sand ratio, and initial void ratio) on the triaxial shear behaviors of cemented sand (i.e., the stress–strain relationship, failure pattern, and evolution of bond breakage) were systematically investigated using the numerical technique of the discrete element method, owing to the advantages of this method over the laboratory test. In the remainder of the paper, details of the DEM simulation, including the elements used in the simulation and the simulation process of the triaxial shearing test, are described first. Then, the shear behaviors in different samples, including the stress–strain response, failure pattern, and bond breakage rate, are compared and discussed to understand the influences of the sampling factors.

## 2. Details of DEM Simulation

Simulations by means of the discrete element method were carried out to produce the shear behaviors of cemented sands with different sampling conditions, which were subjected to the consolidated drained shearing test. Details of the DEM simulations, including the elements used in the simulation and the simulation process, are described below.

### 2.1. Elements Used in the Simulation

The laboratory tests on cemented sand by Wang and Leung [35] served as the reference to the DEM simulations here. In the laboratory tests, samples were prepared by mixing the Portland cement and Ottawa 20–30 sand, and they were enclosed laterally by a flexible membrane and vertically by rigid walls. In order to reproduce these configurations, each DEM simulation used the elements of rigid walls and three kinds of particles, i.e., sand particles, cement particles, and membrane particles.

Sand particles in the samples followed the size distribution in Figure 1, which was scaled up from that of Ottawa 20–30 sand by 3.0 times in order to make the computational cost affordable [35]. As shown in Table 1, the particle density was set as 2650 kg/m^3^, the same as that of Ottawa 20–30 sand. A linear contact model was assigned between the sand particles for its simplicity, following the practice in Guo and Zhao [36]. Therefore, the interparticle contact stiffness was linearized using Equation (1), which was based on the Hertz–Mindlin theory and derived by Li et al. [26].
(1)$$k_{n}=R\cdot {\left(\frac{2\sqrt{2\pi }}{3}\frac{\sqrt{p_{c}}G}{1-\upsilon }\right)}^{2/3}\;\mathrm{and}\;\frac{k_{s}}{k_{n}}=\frac{2\left(1-\upsilon \right)}{2-\upsilon },$$
where *k_n_* and *k_s_* are the contact normal and tangential stiffness, respectively, *p_c_* is the confining pressure and was set as 50 kPa for all the samples, *R* is the radius of the soil particle and was evaluated as 0.8 mm, i.e., half of *D*_50_ in Figure 1, and *G* and *υ* are the shear modulus and Poisson’s ratio of soil particles, respectively; the measured values of quartz crystal with *G* = 29 GPa and *υ* = 0.2 were used here [37]. Accordingly, the contact normal and tangential stiffnesses were estimated in a linear fashion as 5.0 × 10^5^ and 4.0 × 10^5^ N/m, respectively. The coefficient of friction was given as 0.5, which is also the measured value of quartz crystal [38].

Cement particles were generated around the sand particles, and the parallel bond model was assigned between them to form the cementation in the sample. The radii of the cement particle and cementation bond were set as 1/4 of *D*_50_ of the sand particles, in accordance to the X-ray tomography image of cemented sand [39,40]. As can be seen in Table 1, the particle density was given as 3150 kg/m^3^, i.e., the density of Portland cement. Using these configurations, the real sample could be better reproduced in DEM; that is, different amounts of cementation could be controlled by changing the number of cemented particles, and different curing times could be represented by changing the bond strength. For instance, as Portland cement was simulated here, the bond strengths of 5.0 MPa, 2.5 MPa, and 1.25 MPa could be used to represent the curing times of 28 days, 7 days, and 3 days, respectively [41]. Then, stiffnesses for these three kinds of bonds were determined as 20.5, 31.1, and 82.1 GPa/m, respectively; hence, all bonds broke with a tensile displacement of ~60 μm in the simulation, which is in line with the stretching test results on the cemented bonds with various thicknesses by Jiang et al. [42].

Membrane particles were assembled and linked together by the contact bond model to intimate a thin wall-like membrane boundary in DEM. This particle-based membrane boundary was then used to laterally wrap the cemented sample, i.e., the mixture of sand particles and cemented particles. Particle density was given as 1800 kg/m^3^, according to the experimentally measured value by Li et al. [43]. Particle size was set as moderate (~1.0 mm) to eliminate the membrane penetration [44]. Contact bond stiffness was assigned as 2.5 × 10^3^ N/m, i.e., 1/200 of the contact stiffness of sand particles, which could eliminate the lateral constraint on the sample during triaxial test [45].

In addition to the membrane particles, rigid walls were used to vertically wrap the sample, which was frictionless to provide the free end boundary condition. Accordingly, the constraint from the rigid wall was also eliminated, and the strain localization or shear band could develop more freely in the sample [46,47,48,49]. The wall stiffness was identical to that of the sand particle, plausibly turning the particle–wall contact into an interparticle contact.

### 2.2. Simulation Process

The DEM simulation started with the preparation of the cemented sand samples, as shown in Figure 2a. Similar to the practice in the laboratory test, the sand particles and cemented particles were first generated in a rigid mold with a height of 14 cm and a radius of 3.5 cm. The total number of these two kinds of particles was more than 20,000, enabling the sample to serve as a representative element volume and produce the stable shear behavior, according to the simulation results in Kuhn and Bagi [50]. Note that the mold for sample preparation was essentially the combination of a cylindrical rigid wall as the lateral edge and two flat rigid walls as the top and bottom edges. The pressure acting from these rigid walls onto the mixture of sand particles and cemented particles was then adjusted to around 0 kPa. The parallel bond model with the parameters summarized in Table 1 was assigned to the contacts between the two kinds of particles, thereby forming the cemented sand sample.

Since this study was targeted at the influences of three sampling factors (i.e., curing time, cement–sand ratio, and initial void ratio) on the shear behavior of cemented sand, seven different samples were prepared, as summarized in Table 2. Specifically, Samples C28_R5L, C7_R5L, and C3_R5L, which had the same cement–sand ratio of 5% and the same initial void ratio of 0.659, but different bond strength varying from 1.25 to 5.0 MPa, were prepared to investigate the influence of curing time. Note that, as aforementioned, the value of 5.0 MPa could represent a curing time of 28 days for Portland cement, while 2.5 MPa and 1.25 MPa could represent curing times of 7 days and 3 days, respectively. Then, in terms of the cement–sand ratio, Sample C28_R5L served as the reference, and Samples C28_RS3L and C28_R1L were further generated using different numbers of cement particles. Lastly, C28_R5M and C28_R5D were also prepared in order to unravel the influence of initial void ratio on the shear behavior of cemented sand.

After sample preparation in the mold, the rigid lateral wall was removed and replaced by the particle-based membrane, as shown in Figure 2b. To eliminate the leakage of soil particles from the pores of the membrane, the membrane particles were arranged with a hexagon-like pattern, i.e., having each one connected with six neighbors. Details of the establishment process were previously described by Li et al. [45]. Note that the top and bottom ends of the membrane were allowed to deform, which is different to the laboratory setting where these two ends are fastened by O-rings. The purpose was again to reduce the constraint from the boundary condition. Thereafter, with the establishment of the membrane boundary, all samples were consolidated under the confining pressure of 50 kPa, as exemplified in Figure 2c. The confining pressure acting from the rigid wall boundary was realized using the servo-control algorithm proposed by Thornton [51], while that from the membrane boundary was applied using the algorithm proposed by Li et al. [26]. Note that, at the end of consolidation, the void ratio of the sample was measured, taken as the initial value in Table 1.

Finally, the triaxial compression tests were carried out. Samples were sheared under the drained condition by moving the top and bottom walls at a constant rate of 10^−3^ m/s, as indicated by Figure 2d. With this rate, the inertia number of each sample was maintained at a level far lower than 10^−3^, demonstrating that the quasi-static condition was well preserved throughout the test. The shearing process was stopped when the axial strain reached 20%. The stress–strain response, failure pattern, and bond breakage rate in the seven prepared samples were measured. Accordingly, in the next section, the influences of the sampling factors (i.e., the curing time, cement–sand ratio, and initial void ratio) on these shearing responses are discussed.

## 3. Simulation Results and Discussion

### 3.1. Influence of Curing Time

In order to investigate the influence of the curing time (essentially the bond strength), the shearing responses of Samples C3_R5L, C7_R5L, and C28_R5L are compared in this section. Figure 3 shows the stress–strain curves of these three samples. Obviously, both the peak strength and the pre-peak stiffness could increase with the curing time. The strain-softening phenomenon could be observed in Sample C3_R5L (see the orange solid line in Figure 3), which was cured for only 3 days, and the peak stress of 117 kPa was located at the axial strain of 0.96%. Then, for Sample C7_R5L, the peak stress was 178 kPa and the corresponding axial strain was 1.20%, while these two variables for Sample C28_R5L were 270 kPa and 1.50%, respectively (see the green and black solid lines). A similar effect of curing time on the stress–strain relationship was observed in the laboratory results by Li et al. [26]. Comparing the values of these two variables here and those in Table 2, it can be confirmed that the regulation of bond strength to peak strength exists and is linear to some extent. Similar to the peak strength, the residual strength at the axial strain of 20% could also be enhanced by the increase in curing time, and the associated values of this variable are given in Table 3. All the above improvements by the curing time indicate that stronger bonds can store more energy and can better resist the force-chain buckling, thereby developing a higher level of stress in the sample [52,53,54].

In addition to the soil strength, the volumetric response is affected significantly by the curing time. In Sample C3_R5L, although strain softening occurred, the sample volume still contracted (see the orange dashed line). Note that, after the peak, the sample intended to dilate but failed eventually. The reason for this evolution pattern is that the bonds in Sample C3_R5L were weak and were consumed during shearing. According to Kuhn and Bagi [55], the rigid-body rotation of particle pairs dominates the dilation. Then, owing to the weak bonds being gradually consumed, the particle pairs in the sample might be loose and cannot contribute to dilation effectively and persistently. From this viewpoint, stronger bonds may result in a greater tendency of volumetric dilation. As expected, Sample C7_R5L began to dilate slightly by an ultimate value of −0.43%, while Sample C28_R5L dilated more distinctly by −1.65% (see the green and black dashed lines). Clearly, these observations demonstrate that the influence of the curing time/bond strength on the stress–strain response is complicated, which may pose difficulties in the constitutive modeling of cemented sand.

In addition to the stress–strain response, the curing time plays an important role in the failure pattern. Figure 4 displays the appearances of the three samples at the end of the shearing test. As shown in Figure 4a of Sample C3_R5L with a strain-softening but contractive behavior, the deformation was homogenous. Note that, as mentioned in Section 2.2, the top and bottom ends of the membrane were not fixed to eliminate the constraint from the boundary. It is suggested that, if these two ends were fastened, a bulging failure pattern, which was observed in the laboratory test with the early-age sample by [26], would have been reproduced in Sample C3_R5L. Instead, strain localization was observed of Sample C7_R5L, attributed to the stronger bonds (see Figure 4b). In Sample C28_R5L with a strain-softening and dilative behavior, a shear band near the bottom was further generated, as shown in Figure 4c. This is because the breakage of a strong bond is comparable to the force-chain buckling to some extent, which can initialize the shear banding, according to Tordesillas [52].

### 3.2. Influence of Cement–Sand Ratio

Responses of Samples C28_R1L, C28_R3L, and C28_R5L are discussed in this section to understand the influence of the cement–sand ratio on the shear behavior of the cemented sand. Figure 5 depicts the stress–strain responses of these three samples. Similar to the effect of increasing the curing time, the peak/residual strength and the pre-peak stiffness could also be enhanced by increasing the cement–sand ratio. In Sample C28_R1L with a ratio of 1%, although the cement was cured for 28 days, a strain-hardening and contractive behavior seemingly appeared, with a peak stress of only 60 kPa; that is, little improvement was produced by the cementation (see the orange lines in Figure 5). The same observation was also found in the laboratory tests of Li et al. [11]. Then, upon increasing the cement–sand ratio from 1% to 3% in Sample C28_R3L, the bond number was increased from 5246 to 17,106, as given in Table 2. Due to this fundamental change at the microscale, the soil response was improved notably, and strain hardening and volumetric dilation were generated (see the green lines). As the cement–sand ratio increased to 5% in Sample C28_R5L, the bond number was increased to 16,141 and the soil response was further improved, manifested by a higher peak/residual strength and volumetric dilation (see the black lines). Note that the specific values of the soil strengths and volumetric strain in these three samples are summarized in Table 3.

Figure 6 presents the failure patterns of the three samples at the axial strain of 20%. Undoubtedly, the pattern was impacted by the cement–sand ratio. Although Sample C28_R1L exhibited a strain-hardening and contractive behavior, strain localization emerged and a shear band seemingly developed, as shown in Figure 6a. Note that the expected result is similar to that of Sample C3_R5L in Figure 4a, where the deformation was homogeneous. Essentially, even though 1% cement was used in the sand, the breakage of strong bonds could still occur through force-chain buckling, resulting in strain localization. As the cement–sand ratio was increased to 3% in Sample C28_R3L, the shear band became discernable in Figure 6b, in line with its strain-hardening and dilative behavior. As the ratio was further increased to 5% in Sample C28_R5L, the shear band became much more distinct in Figure 6c. Therefore, unlike the curing time/bond strength, the influence of cement–sand ratio is complex, related to the aspect of failure pattern, instead of the stress–strain response. This is due to the effect of bond breakage on localization at the microscale, which cannot be revealed by most constitutive models.

### 3.3. Influence of Initial Void Ratio

To unravel the influence of the initial void ratio, the shear behaviors of Samples C28_R5L, C28_R5M, and C28_R5D are compared in this section. Figure 7 shows the stress–strain responses of these three samples, whereby all samples generated the strain–softening behavior (see all the solid lines). Obviously, with the decrease in the initial void ratio, the peak strength and the pre-peak stiffness could be enhanced. As mentioned in Section 3.1, Sample C28_R5L with an initial void ratio of 0.659 exhibited a peak strength of 178 kPa. In Sample C28_R5M with the initial void ratio decreased by 20% to 0.618, the peak strength was significantly increased to 331 kPa, double that in Sample C28_R5L, while that in Sample C28_R5D further increased to 575 kPa. All these values are summarized in Table 3. Comparing the results in Table 3, it can be found that the peak strength was more sensitive to the initial void ratio than the curing time and cement–sand ratio. However, unlike the peak strength, the residual strength at the end of the shearing test was almost unchanged, i.e., independent of the initial void ratio. This is because the bonds were destroyed in the shearing interface, converting the cemented sand back to be a pure mixture of cement particles and sand particles, which then behaved following the critical state theory.

In line with the strain-softening behavior, all samples exhibited dilative behavior (see the dashed lines). The consistency among these samples implies that once sufficient cement is used and well cured, e.g., using the curing time of 28 days and the cement–sand ratio of 5%, an improvement in the soil, whether loose or dense, can be guaranteed using Portland cement. In contrast, when using an insufficient curing time, e.g., Sample C3_R5L, or insufficient among of cement, e.g., Sample C28_R1L, contractive and strain-hardening behavior may occur. Nevertheless, as the initial void ratio decreased, the volumetric dilation could be further enhanced, and the ultimate values for Samples C28_R5L, C28_R5M, and C28_R5D were −1.65%, −2.70%, and −5.96%, respectively.

Figure 8 shows the failure mode of the three samples. Again, response consistency could be observed among the samples; that is, the shear band persisted near the bottom of each sample. Nevertheless, more soil was involved in the rigid bottom part in Samples C28_R5M and C28_R5L than in Sample C28_R5L, manifested by the shearing interface inclined at a higher angle. To some extent, such a difference in the failure mode is comparable to that in the case of the rigid footing penetrating into the soil foundation; that is, punching failure occurs in the loose sand, which is similar to Figure 8a, and general shear failure occurs in dense sand, which is similar to Figure 8b,c [56].

### 3.4. Accumulative Bond Breakage

In order to provide more insight into the constitutive modeling of cemented sand, the evolutions of bond breakage for the seven samples in Table 2 are also discussed. Here, the rate of accumulative bond breakage Br is defined in Equation (2) and used in the discussion.
(2)$$B_{r}=\frac{N_{b}}{N_{t}}\times 100\%,$$
where *N_b_* is the number of accumulative broken bonds, and *N_t_* is the total number of bonds before shearing. Note that *N_t_* for each sample is given in Table 2, and *N_b_* was measured during the shearing process.

Figure 9 plots the evolutions of *Br* for the seven samples, allowing the influences of different sampling factors to be recognized. Firstly, with the increase in curing time/bond strength, bond breakage may emerge at a greater axial strain and *B_r_* can be lowered throughout the shearing process. To be specific, at the axial strain of 5%, *B_r_* in Sample C3_R5L reached 19.6% (see the cyan line), while the values in Sample C7_R5L and C28_R5L were reduced to 13.9% and 10.5%, respectively (see the earth-yellow and black lines). Secondly, similar to the curing time, by increasing the cement-sand ratio, Br could also be lowered. Here, at the beginning of the test, the evolution patterns among Samples C28_R1L, C28_R3L, and C28_R5L were similar. However, after the axial strain of 3%, the difference grew. At the end of shearing, Br developed to 44.3%, 26.2%, and 21.5% for these three samples, respectively (see the blue, red, and black lines). Lastly, the evolution of Br appeared to be insensitive to the initial void ratio, since the responses of Samples C28_R5L, C28_R5M, and C28_R5D were similar to each other.

## 4. Conclusions

The influences of the sampling factors (i.e., the curing time, cement–sand ratio, and initial void ratio) on the triaxial shear behavior of cemented sand were investigated using DEM simulations. The salient findings are as follows:Peak strength, residual strength, and pre-peak stiffness were enhanced by either increasing the curing time or increasing the cement–sand ratio. The enhancements were fundamentally attributed to the increases in bond strength and bond number.Curing time complicated the stress–strain relationship of cemented sand, since strain-softening but contractive behavior was generated in the sample with a curing time of 3 days. Cement–sand ratio disrupted the correlation between the failure pattern and stress–strain evolution pattern, since the shear band occurred in the sample with strain-softening and contractive behavior, which had a cement–sand ratio of only 1%.By decreasing the initial void ratio, the peak strength and pre-peak stiffness can be significantly enhanced, and the shear band may incline at a higher angle. However, the residual strength and failure pattern are insensitive to this change.Bond breakage may emerge later and be less intensive when increasing the curing time. It can also be intensified due to the medium shearing strain by increasing the cement–sand ratio. However, the whole evolution pattern is insensitive to the change in the initial void ratio.Overall, the mechanical behaviors of cemented sand, in terms of the strength, stiffness, and volumetric dilation, were found to be significantly enhanced by increasing the curing time, cement–sand ratio, and packing density. The failure pattern was also changed, attributed to the regulation of the bond breakage at the microscale. These results provide important insight into other cementation methods, such as using gypsum, biopolymer, or MICP.

## Figures and Tables

**Figure 1 materials-15-03337-f001:**
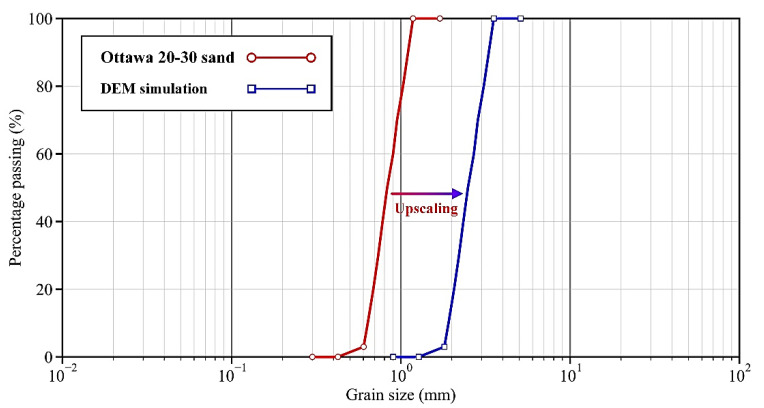
Grain size distribution of sand particles.

**Figure 2 materials-15-03337-f002:**
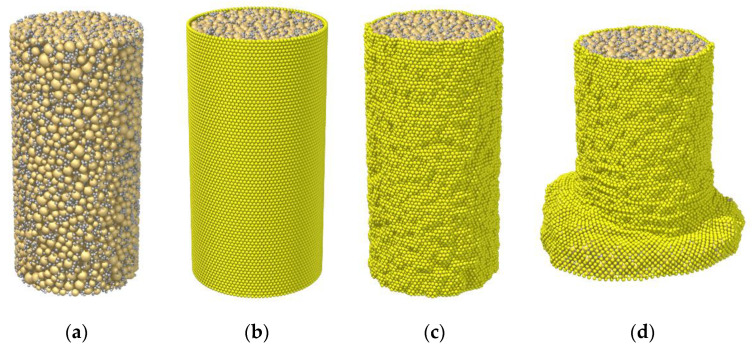
Simulation process: (**a**) sample preparation; (**b**) membrane boundary establishment; (**c**) consolidation; (**d**) triaxial compression test.

**Figure 3 materials-15-03337-f003:**
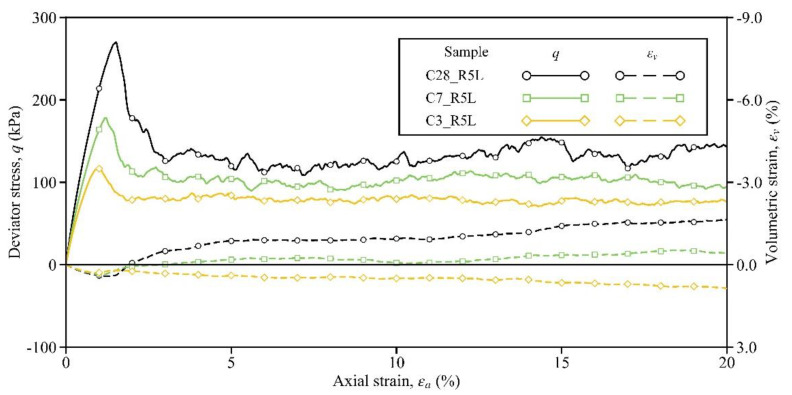
Influence of the curing time on the stress–strain response.

**Figure 4 materials-15-03337-f004:**
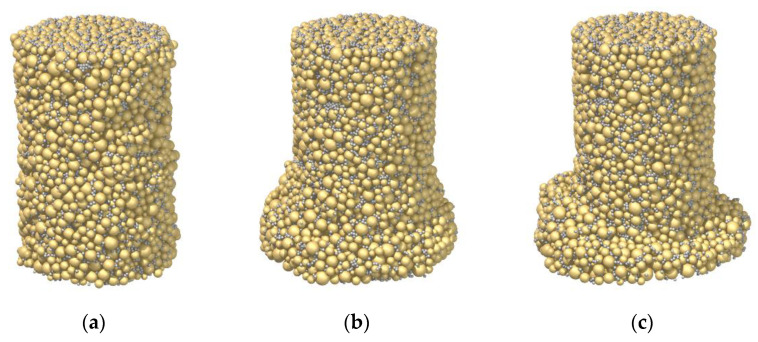
Influence of the curing time on the failure mode: (**a**) Sample C3_R5L; (**b**) Sample C7_R5L; (**c**) Sample C28_R5L.

**Figure 5 materials-15-03337-f005:**
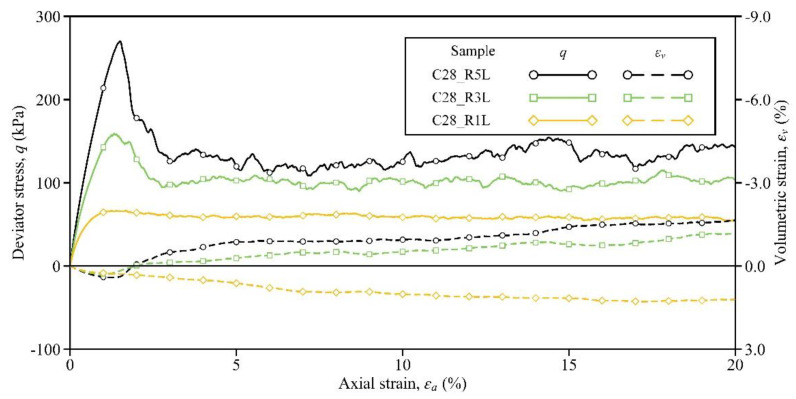
Influence of the cement–sand ratio on the stress–strain response.

**Figure 6 materials-15-03337-f006:**
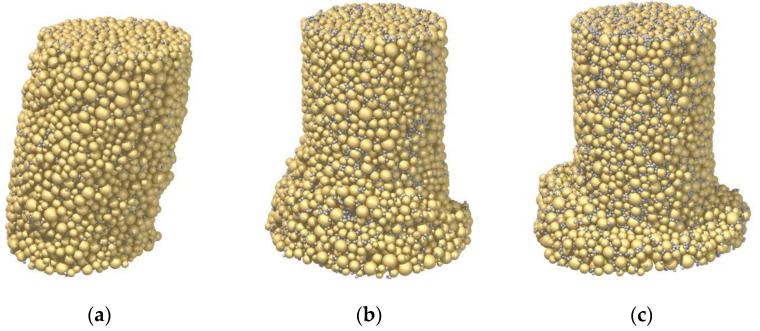
Influence of the cement–sand ratio on the failure mode: (**a**) Sample C28_R1L; (**b**) Sample C28_R3L; (**c**) Sample C28_R5L.

**Figure 7 materials-15-03337-f007:**
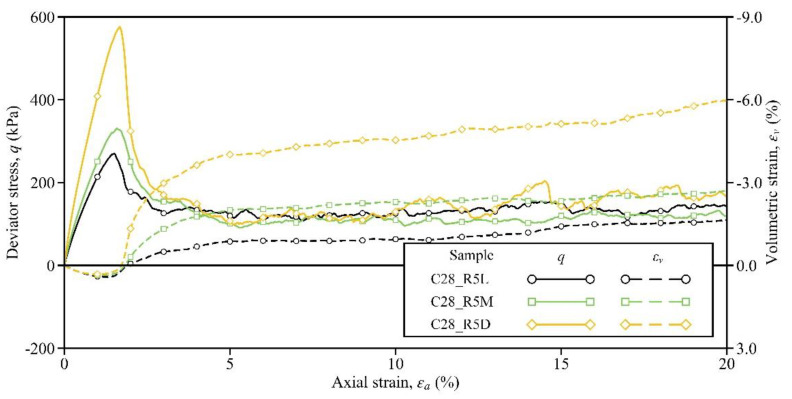
Influence of the initial void ratio on the stress–strain response.

**Figure 8 materials-15-03337-f008:**
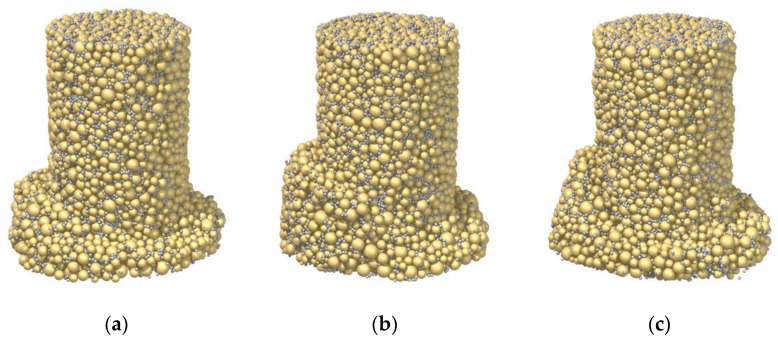
Influence of the initial void ratio on the failure mode: (**a**) Sample C28_R5L; (**b**) Sample C28_R5M; (**c**) Sample C28_R5D.

**Figure 9 materials-15-03337-f009:**
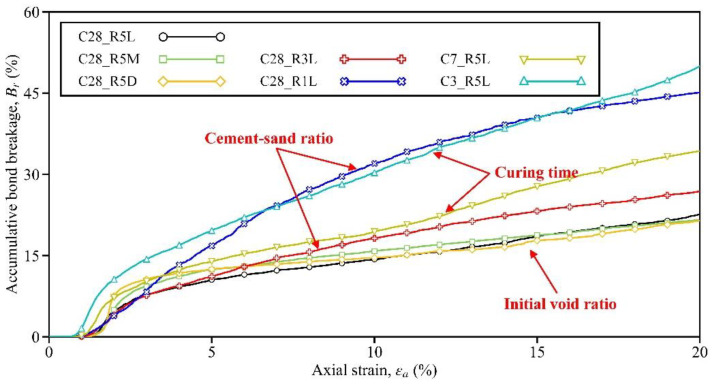
The rate of accumulative bond breakage for the seven samples.

**Table 1 materials-15-03337-t001:** Parameters of three types of particles and rigid wall for all samples.

Elements	Parameters	Values	
**Sand particles**	Density	2650	kg/m^3^
	Particle radius	0.9–3.54	mm
	Contact normal stiffness	5 × 10^5^	N/m
	Contact tangential stiffness	4 × 10^5^	N/m
	Coefficient of friction	0.5	
**Cement particle**	Density	3150	kg/m^3^
	Particle radius	0.62	mm
	Coefficient of friction	0.5	
	Bond radius	0.62	mm
	Parallel bond strength	1.25–5.0	MPa
	Parallel bond stiffness	20.5–82.1	GPa/m
**Membrane particles**	Density	1800	kg/m^3^
	Particle radius	1	mm
	Contact bond stiffness	2.5 × 10^3^	N/m
	Coefficient of friction	0.0	
**Rigid walls**	Normal stiffness	5 × 10^5^	N/m
	Coefficient of friction	0.0	

**Table 2 materials-15-03337-t002:** Parameters of the samples with different initial conditions.

Sample Label	Curing Time	Bond Strength	Cement-Sand Ratio	Soil Particle Number	Cement Particle Number	Bond Number	Initial Void Ratio
	(Days)	(MPa)	(%)	(-)	(-)	(-)	(-)
C28_R5L	28	5.0	5.0	5611	16,141	30,463	6.59 × 10^−1^
C7_R5L	7	2.5	5.0	5611	16,141	30,463	6.59 × 10^−1^
C3_R5L	3	1.25	5.0	5611	16,141	30,463	6.59 × 10^−1^
C28_R3L	28	5.0	3.0	5705	9843	17,106	6.80 × 10^−1^
C28_R1L	28	5.0	1.0	5798	3335	5246	7.00 × 10^−1^
C28_R5M	28	5.0	5.0	5611	16,141	31,063	6.19 × 10^−1^
C28_R5D	28	5.0	5.0	5611	16,141	32,204	4.87 × 10^−1^

C: curing time; R: cement–sand ratio; L/M/D: loose/medium dense/dense sample.

**Table 3 materials-15-03337-t003:** Responses of the samples with different initial conditions.

Sample Label	Peak Deviator Stress	Axial Strain of the Peak	Residual Stress	Ultimate Volumetric Strain
	(kPa)	(%)	(kPa)	(%)
C28_R5L	270	1.50	143	−1.65
C7_R5L	178	1.20	95	−0.43
C3_R5L	117	0.96	77	0.86
C28_R3L	159	1.33	103	−1.15
C28_R1L	60	1.04	54	1.20
C28_R5M	331	1.59	120	−2.70
C28_R5D	575	1.67	169	−5.96

## Data Availability

Not applicable.

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
