# Peer review of "Numerical Investigation of Triaxial Shear Behaviors of Cemented Sands with Different Sampling Conditions Using Discrete Element Method"

_materials, 2022, doi:10.3390/ma15093337_

Round 1
Reviewer 1 Report
Attractive manuscript related to the evaluation of the influence of curing time, cement-sand ratio and initial void ratio on triaxial shear tests using a discrete element method.
In general, the paper is well presented, and only minor details may be clarified or corrected/completed:
- Please explain why you chose these three factors (curing time, cement-sand ratio and initial void ratio);
- Equation (1): please review the expression layout for kn;
- Figure 1: “Percentage finer (%)” or “Percentage passing"?
- Table 1: please include the relevant standards used in the evaluation of the indicated parameters;
- Line 114: “curing time” or “curing times”?
- Line 161: “the lateral rigid wall” or “the rigid lateral wall”?
- Line 199: “curing time indicates” or “curing time indicate”?
- Line 216: “these observations demonstrates” or “these observations demonstrate”?
- Figure 7: the legend of this graph is hiding a part of the “C28_R5L” sample line (ev);
- Line 301: “in the bottom rigid part” or “in the rigid bottom part”?
- Lines 319/320: this information must be placed before Figure 9;
- In section 4, more findings than conclusions are presented. I suggest that, in addition to the findings and respective explanations, you also add (in each paragraph listed) the inductions and inferences that can be pointed out for the mechanical behaviour (or other parameters) of these cemented sands;
- Line 348: “and less intensive” or “and be less intensive”?
- In “References”, you can include the digital object identifier (DOI) for all references where available (as “encouraged” in the “Instructions for Authors”).
Reviewer 2 Report
An interesting work, which is DEM analysis of the sampling factors on the triaxial shear behaviors of cemented sand.
Graphics quality is impressive.
The text is full of references to previous works of the authors. Really 24, 48, etc. does it have to be listed here?
Needs a stylistic check. 2 sentences in a row starting with: "Note that... " (Lines 149-151).
An explanation of the designation of samples could be presented more clearly, for example: C28_R5D, С28 - is for curing time in days, R5 - is for cement-sand ratio in %, D - is for void ratio, dense sample. By the way, what is M? medium by void ratio?
The exact amount of soil particle number or cement particle number is impressive in Table 2 . It would be interesting if previous laboratory studies were briefly highlighted. Especially the results in Table 3, are they presented as a result of simulations only or are they justified somehow by laboratory tests?
Reviewer 3 Report
The reviewed work presents the influence of curing time, cement-sand ratio and initial void ratio on the triaxial shear behaviors of cemented sand. The study was carried out using numerical technique of discrete element method. The work is well written. The authors discussed the measurement results in a clear and concise way. In my opinion, the presented results of numerical calculations are an important supplement to the experimental study. The results presented in the paper may be of interest to the construction community. While reading the work, I did not notice any errors or fragments of the text requiring correction. For these reasons, I propose that the manuscript can be accepted for publication without corrections.
Scientific soundness - I have no reason to question scientific soundness. The literature analysis was carried out reliably. The authors in the "intoduction" summary inform that similar experimental studies have been conducted before. The detailed description of the simulation is provided.
Novelts - Similar works based on numerical simulations were previously published. There is a particular similarity to the work [23] (Li, Z .; Wang, Y.H .; Ma, C.H .; Mok, C.M.B., 2017). The authors of the work [23] analyzed the effect of curing time, while the tested samples did not differ in composition. In the reviewed work, apart from the curing time, calculations were performed for samples with different Cement-sand ratios and different initial void ratio. For these reasons, I believe that there are elements of novelty in the reviewed paper.
Overall merit / flaw. The work presented for review is based only on numerical simulations. The authors did not carry out their own experimental studies that could confirm the correctness of the performed calculations and interpretation of the results. In the discussion, the authors refer to previously conducted research by other authors. The lack of own experimental research always raises doubts as to the correctness of the interpretation of the results. Some journals refuse to publish papers based solely on numerical simulation. However, it is not my role to judge whether such work should be published in Materials. It depends on the rules of the journal. English level. In my opinion, the level of English is acceptable, although the paper requires some linguistic correction.
Author Response
The authors are thankful for the reviewer's comment. The quality of the manuscript has been further improved following the reviewer's suggestion.